# Perioperative management of kidney transplantation in China: A national survey in 2021

Ziyu Zhu[1]☯, Xiaoying Chi[1]☯, Yuwen Chen[1], Xiaowen Ma[1], Ying Tang[1], Dawei Li[2], Ming Zhang[2]*, Diansan Su📵[1]*

1 Department of Anaesthesiology, Renji Hospital, School of Medicine, Shanghai Jiaotong University, Shanghai, China, 2 Department of Urology, Renji Hospital, School of Medicine, Shanghai Jiao Tong University, Shanghai, China

☯ These authors contributed equally to this work.
* drmingzhang@126.com (MZ); diansansu@yahoo.com (DS)

**Data Availability Statement:** All relevant data are within the paper and its Supporting Information files.

**Funding:** This study was supported by the National Natural Science Foundation of China (82001227,

## Abstract

Perioperative anaesthesia management has an important significance for kidney transplantation; however, the related consensus remains limited. An electronic survey with 44 questions was developed and sent to the chief anaesthesiologist at 115 non-military medical centres performing kidney transplantation in China through WeChat. A response rate of 81.7% was achieved from 94 of 115 non-military medical centres, where 94.4% of kidney transplants (10404 /11026) were completed in 2021. The result showed an overview of perioperative practice for kidney transplantations in China, identify the heterogeneity, and provide evidence for improving perioperative management of kidney transplantation. Some controversial therapy, such as hydroxyethyl starch, are still widely used, while some recommended methods are not widely available. More efforts on fluid management, hemodynamical monitoring, perioperative anaesthetics, and postoperative pain control are needed to improve the outcomes. Evidence-based guidelines for standardizing clinical practice are needed.

## Introduction

Kidney transplantation is the preferred treatment for end-stage renal disease (ESRD), which emerged from advanced and irreversible chronic kidney disease (CKD). China had the most patients with CKD (132 million), accounting for nearly one-fifth of the global total [1]. ESRD affects more than 1,500 per 1 million population in countries such as the United States and Japan [2]. One-fourth of persons with ESRD receive kidney transplants[3].

Although the survival rate of kidney transplantation has increased significantly in recent years, the risk of perioperative complications remains, such as acute kidney injury (AKI), delayed graft function (DGF), infection, hemorrhage, and venous thrombosis. Approximately 25% of all kidney recipients suffer from postoperative DGF, needing renal replacement therapy, resulting in an increase in mortality of 40% [4].

82371193, U21A20357, 81970995), Science and Technology Commission of Shanghai Municipality Funding (21S31900100), the Incubating Program for Clinical Research and Innovation of Renji Hospital(RJTJ-JX-002, RJPY-DZX-007, PYII20-09), Shanghai municipal Education Commission-Gaofeng Clinical Medicine Support (20191903), and Shanghai Municipal Health Commission Key Support Project (2023ZDFC0201).

**Competing interests:** The authors have declared that no competing interests exist.

**Abbreviations:** ESRD, end-stage renal disease; CKD, chronic kidney disease; AKI, acute kidney injury; DGF, delayed graft function; CD, cadaveric donor; ECG, Electrocardiography; TTE, transthoracic echocardiography; HES, hydroxyethyl starch; CVP, central venous pressure; MAP, mean arterial pressure; SBP, systolic blood pressure; CVC, central venous catheter; CO, cardiac output; TEE, transesophageal echocardiography; PCIA, Patient-controlled intravenous analgesia; NSAIDs, non-steroidal anti-inflammatory drugs; EAU, European Urology; LRKT, living-related kidney transplantation; ASA, American Society of Anaesthesiologists.

With the growing use of marginal organs for renal transplantation and aging of donors and recipients, optimized perioperative management to improve the renal outcome has become a new challenge. Previous studies have shown that optimal fluid therapy could decrease DGF after renal transplantation [5]. In addition, intraoperative hypotension should be carefully regulated due to an increased risk of AKI in a nontransplant study [6].

Anaesthesiologists play a role in assessing the perioperative status of patients with high-risk conditions, optimizing hemodynamic administration and avoiding potentially nephrotoxic drugs to reduce postoperative complications. Existing guidelines for kidney transplantation mainly focus on surgical aspects [4, 7–10]. However, comprehensive guidelines for anaesthesia-related administration in kidney transplantation are lacking. The investigation of the current clinical status of anaesthesia management has an important significance for improving the prognosis of kidney transplantation.

Survey results from France and UK show little national consensus on anaesthesia management for kidney transplantation [6, 11]. The actual clinical practice remains unknown in China. Thus, we conducted a survey to generate an overview of the current situation of national practice across Chinese renal transplant units to identify heterogeneity and provide evidence for improving perioperative management of kidney transplantation.

## Materials and methods

### Questionnaire design and conduct of survey

Based on the result of the pilot survey conducted from April to May 2022, a final version of the electronic survey with 44 questions was decided and sent to the chief anaesthesiologist through WeChat (Tencent Inc., Shenzhen, China) at each of the 115 non-military medical centres performing kidney transplantation in China from May to September 2022, according to the list of institutions qualified of kidney transplantation published by the National Health Commission of the People's Republic of China on June 11, 2021. The survey covered respondent demographics (4 questions), general situation of renal transplantation (4 questions), preoperative management (4 questions), intraoperative management (28 questions) including anaesthesia drugs and methods (6 questions), fluid management (11 questions), hemodynamic management (11 questions), and postoperative management (4 questions) (S1 File).

Before answering the electronic questionnaire, consent will be informed through a cover letter, which was attached to clarify the purpose, general content, and privacy statement of this survey. The participants were informed that this survey would collect the name, sex, age, and IP address to validate the data they entered. In addition, the cover letter informed the respondents about the data to prepare before completing the questionnaire. They were informed that their data would be stored and their privacy maintained in password-protected computers and that any identifying information would be delinked after analysis and publication. The questions could be answered only after the respondents agreed with this electronic informed consent. Data cleaning was conducted after collecting the results to the greatest extent. All data were carefully checked and confirmed with respondents in case of any doubt.

The official number of kidney transplantations, covering 31 provincial-level administrative regions in mainland China (Hong Kong, Macao, and Taiwan were excluded), was provided by the kidney transplant surgeon who can access the national scientific registry of kidney transplantation (http://www.csrkt.org.cn/).

The survey respondents were anaesthesiologists instead of the patients. Ethical exemption was approved by Ethics Committee of Renji Hospital, School of Medicine, Shanghai Jiaotong University. Consent will be informed through a cover letter on the questionnaire homepage.

Once the name of investigator was filled in the questionnaire, it is considered to obtaine the informed consent.

## Statistical analysis

Means, standard deviations, and 95% confidence intervals were reported for symmetric distributed continuous variables. Median, interquartile range (IQR), minimum and maximum were calculated for nonsymmetric continuous variables. Categorical data were presented as frequency and percentage. All data were collected with Microsoft Access and analysed by Prism 7.0 (GraphPad Software, La Jolla, CA, USA).

## Results

Completed questionnaires were obtained from 94 of 115 non-military medical centres (response rate of 81.7%), where 94.4% of kidney transplants (10404 /11026) were completed in 2021, including 98.4% of living-donor transplantations (2275/2311) and 93.3% of cadaveric donor (CD) transplantations (8129/8715). The total number of CD transplants in 94 non-military centres according to the distribution by provinces in 2021 is shown in Fig 1A. Most of the hospitals that replied (58.5%, 55 centres) performed renal transplantations before 2000, two of which started renal transplantations in 1970 (Fig 1B). From the response rate by provinces, 17 provinces reached 100%, 3 have 50% (1 of 2 centres answered in Heilongjiang, Shanxi, and Xinjiang) and 1 has 0 response (0 of 1 centre in Tibet with 0 renal transplantations completed in 2021). Centres in the south have completed significantly more kidney transplants than those in northern provinces. Further details are shown in S1 Table. Moreover, 84% (79 centres) of the investigated centres established specialized protocol for perioperative anaesthesia management of kidney transplantation, whereas the others 16% (15 centres) did not.

### Preoperative management

Electrocardiography (ECG) is most commonly recommended for preoperative cardiopulmonary system evaluation in patients with ESRD, which is routinely performed by 95.7% of the investigated centres (90/94). The most common preoperative cardiopulmonary assessments included transthoracic echocardiography (TTE), exercise tolerance, and pulmonary function tests (S1A Fig). Preoperative hypertension control (89.4%), blood glucose management (90.4%), and preventive antibiotics (96.8%) are routinely implemented in most centres.

### Anaesthesia management

Propofol is the most popular anaesthetic agent for induction (81%) and maintenance (88%). In addition, 52% of the centres use etomidate, and 55% of the centres use midazolam during induction. Sufentanyl is the most commonly used opioid during induction (93%) and maintenance (50%). Remifentanyl is rarely used for induction but is sometimes chosen for maintenance (35%). Non-depolarizing muscle relaxants applied in 70% of the centres during maintenance, among which cisatracurium is the most commonly used muscle relaxants (84%) for induction (Table 1).

Combined intravenous and inhalation anaesthesia is applied in 93% of the centres (88/94). Most centres provide sevoflurane as inhaled anaesthetic (87%), whereases five centres report using desflurane and one centre uses isoflurane (Table 1).

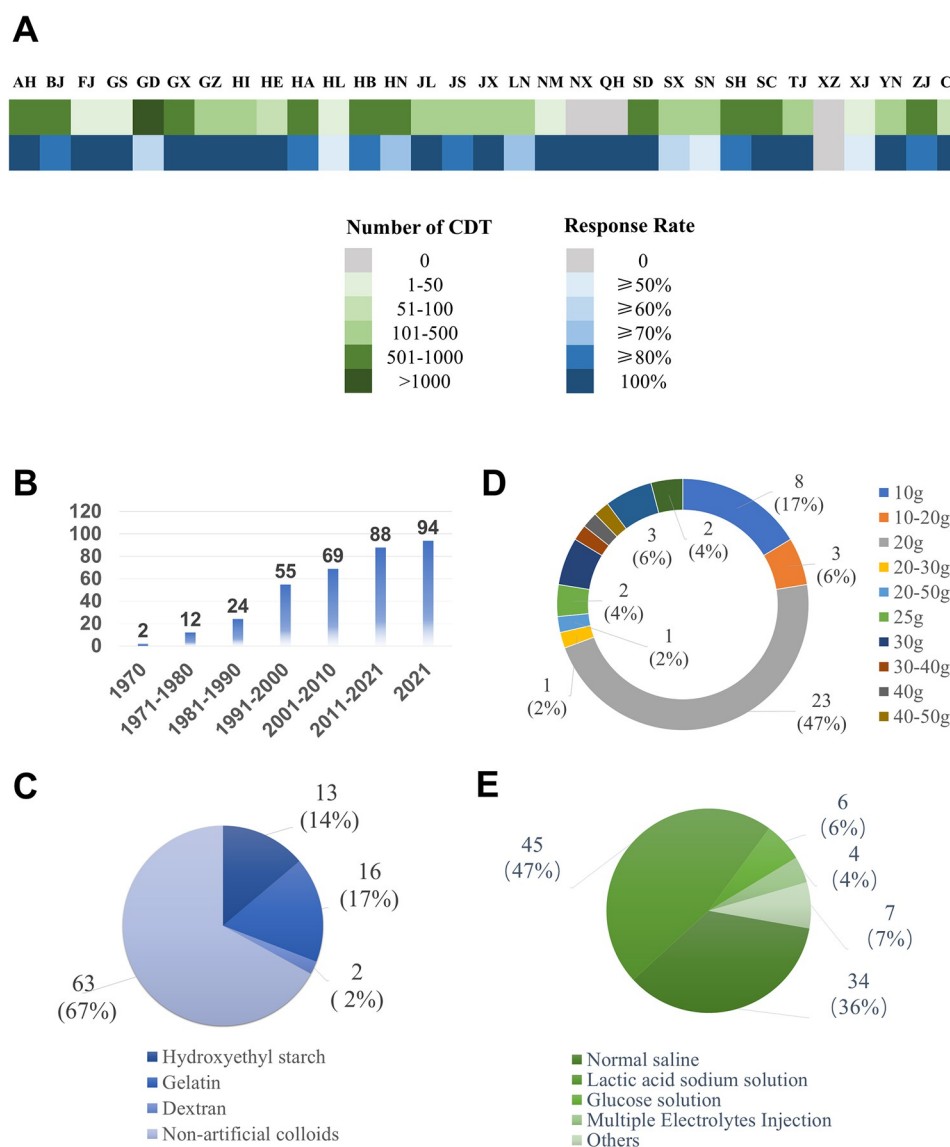

**Fig 1. General situation of investigated centers in China and situation of intraoperative fluid administration.** A, Heat map of the total number of completed kidney transplants and response rate from replied centers in each provinces in China in 2021. Green blocks correspond to the total number of completed cadaveric donor kidney transplants, while blue blocks correspond to the response rates from replied centers. B, Trend chart of the initial years of kidney transplantation. C, Investigation on the use of intraoperative artificial colloids. D, Investigation on the total dose of albumin during kidney transplantation. E, Investigation on the use of intraoperative crystalloids. CDT, cadaveric donor transplantations; AH, Anhui; BJ, Beijing; FJ, Fujian; GS, Gansu; GD, Guangdong; GX, Guangxi; GZ, Guizhou; HI, Hainan; HE, Hebei; HA, Henan; HL, Heilongjiang; HB, Hubei; HN, Hunan; JL, Jilin; JS, Jiangsu; JX, Jiangxi; LN, Liaoning; NM, Inner Mongoria IM; NX, Ningxia; QH, Qinghai; SD, Shandong; SX, Shanxi; SN, Shaanxi; SH, Shanghai; SC, Sichuan; TJ, Tianjing; XZ, Tibet; XJ, Xinjiang; YN, Yunnan; ZJ, Zhejiang; CQ, Chongqing.

## Fluid management

Sixty-three centres do not routinely apply colloids during renal transplantation, where crystalloid is their primary fluid administration. Approximately half of the centres use sodium lactate solution as the primary crystalloid, whereas 34 use 0.9% saline (Fig 1E). In 31 centres who use colloid, 13 centres reported the use of hydroxyethyl starch (HES), whereas 16 would use gelatin and 2 centres use dextrose (Fig 1C). Human albumin is routinely applied in 54 centres (57.4%)

**Table 1. Intravenous anesthetics during the induction and maintenance of anesthesia.**

|  | Induction (%) | Maintenance (%) |
|---|---|---|
| Propofol | 81 (86.2) | 88 (93.6) |
| Etomidate | 52 (55.3) | - |
| Midazolam | 55 (58.5) | - |
| Sufentanyl | 93 (98.9) | 50 (53.2) |
| Fentanyl | 1 (1.1) | 3 (3.2) |
| Remifentanyl | 1 (1.1) | 35 (37.2) |
| Depolarizing Muscle Relaxants | - | 0 |
| •Succinylcholine | 0 | - |
| Non-depolarizing Muscle Relaxants | - | 70 (74.5) |
| •rocuronium | 24 (25.5) | - |
| •Atracurium | 0 | - |
| •Cisatracurium | 84 (89.4) | - |
| •Vecuronium | 1 (1.1) | - |
| Others | 1 (1.1) | 1 (1.1) |
| Inhalation anesthetics (IA) |  |  |
| •Sevoflurane | - | 82 (87.2) |
| •Desflurane | - | 5 (5.3) |
| •Isoflurane | - | 1 (1.1) |
| •Non-IA | - | 6 (6.4) |

The number before parentheses represents the number of centers, while the number inside parentheses represents the percentage.

as intravenous fluid (S1B Fig). Moreover, most of these centres (34/54) normally use 10–20 g of human albumin intraoperatively for one kidney transplantation, whereas two centres use up to 100 g of human albumin (Fig 1D).

Fluid management guided by goal-directed therapy based on the stroke volume variation (90%) and pulse pressure variation (10%) was described in 30 centres. Meanwhile, 33 centres perform fluid therapy according to the central venous pressure (CVP) and 26 centres conduct liquid management based on experience. Other basis for fluid administration includes MAP (1 centre), Vigileo (1 centre), and echography guidance (2 centres).

## Hemodynamic monitoring and management

Invasive arterial blood pressure monitoring is regularly applied in 90% of the centres (85) (S1C Fig). In total, 86 centres target a specific systolic blood pressure (SBP) after renal artery opening, and 50% of the centres (43/86) control SBP within 140–150 mmHg, whereas 23.3% centres (20/86) control it within 150–160 mmHg, with the highest range no more than 160–170 mmHg (S1D Fig).

More than half the centres (59/94) regularly insert a central venous catheter (CVC), among which one centre routinely places the CVC via the subclavian approach, whereas others choose the internal jugular approach. In all of these 59 centres, 31 target a specific CVP intraoperatively, and 67.7% (21 centres) of the centres control the CVP within 10%–12% of the baseline after opening the renal artery (S1E Fig).

Only 11 centres routinely conduct cardiac output (CO) monitoring, whereas 16 centres occasionally perform CO monitoring. Arterial waveform analysis including Flotrac and LiDCO rapid is the most commonly employed CO monitoring method in 21 centres. Seven centres use PICCO and 3 centres use transesophageal echocardiography (TEE) for CO

monitoring. Furthermore, only one centre (1/94) regularly perform TEE during renal transplantation (S1F Fig).

Dopamine is the most commonly employed vasoactive agents in 80 (85.1%) centres. Norepinephrine is commonly used in 41 centres reserved as a secondary agent. Twenty-four centres use ephedrine, and 19 centres use phenylephrine. Only one centre does not employ vasoactive agents in general condition (S2A Fig).

Furosemide as diuretic is routinely used in nearly all centres (90/94). Mannitol is routinely used in 21.3% of the centres, whereas 62.8% of the centre indicated that mannitol was not routinely used. Glucocorticoids are routinely applied in 82 centres (S2B Fig).

Moreover, 62.8% (59/94) of the centres initiate blood transfusion when the hemoglobin concentration is less than 7 g/L, whereas 33.0% (31/94) set the transfusion threshold to 8 g/L, and 10 g/L was chosen by six centres (S2C Fig). Blood loss in renal transplantation ranges from 100 to 250 mL in more than half of the centres (48 centres) and less than 100 mL in 35.1% (33 centres). Intraoperative temperature monitoring is routinely performed in 82 (87.2%) centres.

## Postoperative recovery

Patients are resuscitated and extubated before returning to the relevant ward in 83.0% of the centres (78/94), whereas 14 centres reported that patients are transferred to the ICU intubated after renal transplantation (S2D Fig). During the recovery period, muscle relaxant antagonists are regularly used in 45.7% of the centres (43 units), with the majority of these using neostigmine and atropine (90.7%, 39 units). Sugammadex was routinely used in two centres and occasionally used in two centres when rocuronium was used.

## Postoperative analgesia

Patient-controlled intravenous analgesia (PCIA) is provided postoperatively in most of the centres (87.2%, 82 units), whereas patient-controlled epidural analgesia is used in four centres. Transversus abdominis plane blocks are the most commonly selected regional block method, which is used in 29 (30.9%) centres; others use regional block, including paravertebral block (6 units), lumbar quadratus block (1 unit), and local infiltration anaesthesia (2 units). Three centres reported non-postoperative analgesia after renal transplantation, whereas one centre indicated that the analgesia strategy is determined by the surgeon (S2E Fig). If PCIA is provided, sufentanyl is selected as opioids by 86 centres. Other opioids include fentanyl (3 units), oxycodone (15 units), nabuphine (1 unit), butorphanol (3 units), and hydroxymorphone (5 units). Other intravenous drugs added to the PCIA included non-steroidal anti-inflammatory drugs (NSAIDs) (21 units), tramadol (4 units), ketamine (1 unit), and dexmedetomidine (1 unit) (S2F Fig).

## Discussion

To our knowledge, this is the first nationwide survey on the current status of perioperative management of renal transplantation from the perspective of anaesthesia in mainland China, and the results showed a clear heterogeneity in perioperative anaesthesia practice, and many issues should be resolved.

Fluid management is a major challenge in the anaesthesia administration of renal transplantation. CVP-directed infusion produced a more stable hemodynamic profile, better diuresis, and early graft function, which is recommended by European Urology (EAU) [12]. The principles of fluid management during kidney transplantation vary among centres in China, with only one-third of centres guided by CVP, and another third by experience, although the

CVC was inserted in more than half of the centres (59/94) in China. In France, CVP-guided volume infusion remained the traditional hemodynamic approach [6], whereas the survey from the UK revealed that the utility of CO-based goal-directed fluid administration is still being debated and three-quarters of centres with CVP targeting [11]. Only one-tenth of the investigated centres in our survey (11 centres) routinely conduct CO monitoring, and TEE is regularly used in only one centre during renal transplantation. Although esophageal Doppler and volume variation to guide fluid administration appear to be promising, related reports are still limited [13].

A huge controversy remains in determining whether crystalloids or colloids are better for intravenous fluid management. Some data suggest that HES may be associated with an increased incidence of AKI and decreased survival [14, 15], and its use was reported in 13 centres from our survey. However, the evidence that HES is harmful to postoperative renal function in patients undergoing kidney transplantation or other major surgeries is insufficient. The result of the trial that included eight patients undergoing living-related kidney transplantation (LRKT) indicated that both HES 130/0.4 and 4% succinylated gelatin can be safely used for LRKT and that HES 130/0.4 was associated with a more rapid recovery of renal function [16]. Similarly, the renal function of 109 patients who received renal transplantations in the HES group was comparable with that of the gelatin albumin group [17]. However, avoiding HES in kidney transplant recipients still appears prudent, as recommended by the committee on transplant anaesthesia of the American Society of Anaesthesiologists (ASA) in 2021 [13].

Albumin, routinely applied in 57.4% of the investigated centres in the present survey, is another type of colloid widely used in kidney transplantation, with several theoretical advantages such as increased plasma oncotic pressure, antioxidant effects, anti-inflammatory properties, and positive effects on vessel wall integrity [18]. Most of the centres in the present survey normally use 10–20 g of human albumin intraoperatively for one kidney transplantation and no more than 100 g. Prevous studies have supported albumin use in kidney transplantation; however, recent studies found that albumin is not advantageous over crystalloids alone in kidney transplantation; thus, albumin should be used according to the suggestion from the ASA rather than per protocol [13].

For consideration on the ideal type of crystalloid in kidney transplantation, 0.9% saline was traditionally preferred based on the belief that potassium-containing solutions may aggravate hyperkalemia; however, several studies have found that 0.9% saline is more likely to cause hyperkalemia than balanced crystalloid solutions such as Lactated Ringers and Plasma-Lyte [19, 20]. Recent evidence supports favoring balanced crystalloid solutions over 0.9% saline in kidney transplantation [13]. In the present survey, kidney transplant centres have gradually changed their concepts in China, in which half of the centres preferred sodium lactate solution as a primary crystalloid. However, more efforts should be made because 36.2% of the centres still prefer 0.9% saline. More clinical studies are needed to explore the optimal scheme for the type of fluid or the monitoring measures in kidney transplantation.

The nephrotoxicity of sevoflurane have been widely reported due to the production of "compound A", while sevoflurane was widely used in 87.2% of investigated centers. Compound A is generated as a result of a chemical reaction between sevoflurane and the carbon dioxide absorbent. Some previous studies showed that Compound A harms was considered to have nephrotoxicity in rats[21]. However, more and more evidences have shown no negative effect on renal function[22–24], and sevoflurane can be used safely for renal transplant surgery, which is displayed in some guidances and expert consensus[25].

Low intraoperative arterial pressure during reperfusion appears to be associated with DGF. In the present survey, 91.5% of the centres target a specific SBP after renal artery opening, in which it ranges from 140–150 mmHg in half of the centres. Normally, MAP should be kept

over 80 mmHg; however, no specific MAP value has been established according to the Chinese Anaesthesia technique guideline for renal transplantation (version 2019) by the Chinese Society of Organ Transplantation of the Chinese Medical Association, which just pointed out that the blood pressure should be adjusted not to be lower than the preoperative baseline [25]. However, the SBP should be within 140–160 mmHg before renal artery opening [25].

Dopamine was the most commonly used vasopressor, selected by 85.1% of the centres in China. Although a low dose of dopamine was considered associated with improved urine output and early graft function, the beneficial effects of dopamine on renal function are conflicting [26, 27]. EAU guidelines on renal transplantation prohibit the routine use of low-dose dopaminergic agents [4]. If hypotension is encountered intraoperatively, norepinephrine may be preferred as an agent with beta-adrenergic activity [28, 29], reserved as a secondary agent by 41 Chinese centres. In addition, vasopressors should be carefully considered due to potential renal vasoconstriction [4].

Non-opioid analgesics could be combined with opioids to decrease opioid use and provide better postoperative pain control. Twenty-one Chinese centres use NSAIDs in the PCIA; however, it is recommended to be avoided in kidney transplant recipients because of potential nephrotoxicity [30]. If NSAIDs are used, follow-up testing of graft function and hyperkalemia may be needed [31] since the postoperative use of NSAIDs is associated with a significant increase in the risk of AKI [32]. Regional analgesia should be considered for postoperative analgesia, as it offers significant postoperative pain control, reduced postoperative opioid, and less antiemetic requirements. However, the proportion of regional blockage after kidney transplantation in China is still less than half (40.4%, 38 centres), which may need improvement for anaesthesia management.

Our results showed the controversies and variations in the clinical practice of kidney transplantation in China; however, some limitations exist. First, Chinese military hospitals are not included in our survey because of potential privacy restrictions. Fortunately, the response rate to our survey is quite high, as the survey included all of the top 10 centres in terms of kidney transplant amounts, where 94.4% of kidney transplants were completed in 2021. Second, we have not investigated long-term follow-up status postoperatively. More studies on postoperative complications such as DGF are needed. Third, survey responses may underestimate the clinical reality due to the recall bias in retrospective studies. Furthermore, there are certain deficiency in some optains for certain question, such as Question 28 about the target range of CVP, even though we did a pilot study.

## Conclusion

This study demonstrated the heterogeneity in the current anaesthesia administration of kidney transplantation in China. More efforts are needed to improve outcomes of renal transplantation, including optimized fluid management, avoidance of potential nephrotoxic substances such as HES and NSAIDs, implementation of hemodynamical monitoring, and ideal postoperative pain control. For some controversial issues, evidence-based guidelines must be developed for standardization in clinical practice.

## Supporting information

**S1 Fig. Periperative monitoring and intraoperative fluid administration.** A, Preoperative cardiopulmonary function assessments. The column represents the number of hospitals, and the dot represents the percentage. B, Status of routine use of albumin. C, Investigation on the routinely use of invasive arterial monitor. D, The specific targeted value of intraoperative systolic blood pressure (mmHg) in renal transplant recipients after renal artery opening. E,

Specific target of CVP range (cmH2O) after renal artery opening. F, CO monitoring methods employed during renal transplantation. ET, Exercise tolerance; ECG, electrocardiogram; EC, Echocardiography; PFT, Pulmonary function tests; CPET, cardiopulmonary exercise test; CAG, coronary angiogram; MPS, Myocardial perfusion scan; SBP, systolic blood pressure; CVP, central venous pressure; CO, cardiac output; PICCO, Pulse indicator Continuous Cardiac Output; TEE, transesophageal echocardiography.
(TIF)

**S2 Fig. Intraoperative hemodynamic management and postoperative administration.** A, Most commonly used vasoactive drugs. B, The routinely use of furosemide, Mannitol and Glucocorticoids. C, The limit value of Hemoglobin for starting the transfusion. D, The postoperative destination of renal transplant recipients. E, The main methods of postoperative analgesia after kidney transplantation. F, The most commonly used intravenous analgesics in PCIA. PCIA, patient-controlled intravenous analgesia; PCEA, patient controlled epidural analgesia; TAP, Transversus Abdominis Plane block; PVB, paravertebral block; QLB, lumbar quadratus muscle block; LIA, local infiltration anesthesia; ICU, intensive care unit.
(TIF)

**S1 File. Translation of this questionnaire with containing 44 questions.**
(DOCX)

**S1 Table. Information of replied centers and official centers divided by province in 2021.**
KT, Kidney transplantation.
(DOCX)

## Acknowledgments

The authors would like to thank all the clinicians who took time to respond to this questionnaire.

## Author Contributions

**Conceptualization:** Ziyu Zhu, Dawei Li, Ming Zhang, Diansan Su.

**Data curation:** Ziyu Zhu, Ming Zhang, Diansan Su.

**Formal analysis:** Ziyu Zhu, Xiaoying Chi, Ying Tang.

**Project administration:** Dawei Li, Ming Zhang, Diansan Su.

**Supervision:** Diansan Su.

**Validation:** Ziyu Zhu, Xiaoying Chi, Yuwen Chen, Xiaowen Ma, Ying Tang, Dawei Li, Diansan Su.

**Visualization:** Ming Zhang.

**Writing – original draft:** Ziyu Zhu.

**Writing – review & editing:** Ming Zhang, Diansan Su.

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
