## [Decision Letter · Decision Letter 0]

13 Nov 2023

PONE-D-23-30552Perioperative management of kidney transplantation in China: A national survey in 2021PLOS ONE

Dear Dr. Su,

Thank you for submitting your manuscript to PLOS ONE. After careful consideration, we feel that it has merit but does not fully meet PLOS ONE’s publication criteria as it currently stands. Therefore, we invite you to submit a revised version of the manuscript that addresses the points raised during the review process.

We look forward to receiving your revised manuscript.

Kind regards,

Wan-Jie Gu

Academic Editor

PLOS ONE

4. We note that Figure 1A in your submission contain [map/satellite] images which may be copyrighted. All PLOS content is published under the Creative Commons Attribution License (CC BY 4.0), which means that the manuscript, images, and Supporting Information files will be freely available online, and any third party is permitted to access, download, copy, distribute, and use these materials in any way, even commercially, with proper attribution. For these reasons, we cannot publish previously copyrighted maps or satellite images created using proprietary data, such as Google software (Google Maps, Street View, and Earth). For more information, see our copyright guidelines: http://journals.plos.org/plosone/s/licenses-and-copyright.

1. You may seek permission from the original copyright holder of Figure 1A to publish the content specifically under the CC BY 4.0 license. 

6. We notice that your supplementary figures are uploaded with the file type 'Figure'. Please amend the file type to 'Supporting Information'. Please ensure that each Supporting Information file has a legend listed in the manuscript after the references list.

Reviewers' comments:

Reviewer's Responses to Questions

**Comments to the Author**

1. Is the manuscript technically sound, and do the data support the conclusions?

Reviewer #1: Yes

Reviewer #2: Yes

2. Has the statistical analysis been performed appropriately and rigorously? 

Reviewer #1: Yes

Reviewer #2: Yes

3. Have the authors made all data underlying the findings in their manuscript fully available?

Reviewer #1: Yes

Reviewer #2: Yes

4. Is the manuscript presented in an intelligible fashion and written in standard English?

Reviewer #1: Yes

Reviewer #2: Yes

5. Review Comments to the Author

Reviewer #1: This is a rare study of perioperative anesthesia management for kidney transplantation. This study is very interesting because studies in related fields are rare. The researcher collected basic information on anesthesia management for renal transplantation in China, which is helpful for Chinese anesthesiologists to enhance perioperative anesthesia management and improve the outcome of renal transplantation surgery. However, the article still needs some improvement.1. Please elaborate on the statistical scheme.2. the discussion section deals with the issue of perioperative volume management, which is of interest to many clinicians. Based on the huge data support, is it possible to draw the advantages and disadvantages of perioperative volume management measures? 3. For many perioperative management protocols, there are variations in different clinical centers, and does this variation affect the prognosis of renal transplantation?

Reviewer #2: The manuscript "Perioperative management of kidney transplantation in China: A national survey in 2021" is very well written and structured, and conclusions are supported by high quality data. The importance and novelty of this study compared to published work in this subject area is good. However, there are still some issues needed to be addressed.

1 In this study, the authors applied wechat questionnaire. The response rate is high. Are there any disadvantages or limitations for this kind of electronic online survey?

2 The nephrotoxicity of sevoflurane have been widely reported due to the production of “compound A”. However, most of the centers routinely used sevoflurane during renal transplantaion. This point should be discussed.

3 CVP can be the target of GDFT, just like SVV,PPV,or CO?

4 For questionnaire, there are some typos. For example,

①Could you expand the abbreviation of MCQ ?

②Q3,Q4: If the interviewee is not an anesthesiologist, the questionaire was analysed differently?

③Q15,Q17,Q36,Q42,Q44: “rocu ,[propofol,sufentanyl,fentanyl],adrenaline,[fentanyl , sufentanyl], general medical ward” . Please keep the first letter in capitals.

④Q18:Could you specify the monitor of depth of anesthesia?

⑤Q19：“Goal-directed hemodynamic therapy(GDFT)” and “CVP-guided”, there is some overlap.

⑥Q28：“□<4% □4-6% □7-9% □10-12% □>12%”. If the baseline CVP is 10, the target range should be 0.4，0.4-0.6，0.7-0.9，1-1.2，1.2. However, “0.4，0.4-0.6，0.7-0.9” are not different in clinical practice.

⑦Q31: If the target is intraoperative MBP but not SBP, how to answer this question?

⑧Q35: TTE is non-invasive. TEE is invasive. They are not comparable. Specify should be better.

⑨Q37：What does “Hormone” refer to? Predisone？Methyl-prednisone?Estrogen?

5 For figures, the following questions should be addressed.

①Fig1: the style of figure legend (fig1C, fig1D, and fig1E) is not consistent.

②Fig1C:“non-” N should be in capital.

③Fig1E: multiple electrolytes injection refers to sodium acetate/carbonate Ringer’s injection?

6 Line161 “sufentail”, line162 “remifentanyl”, line 291”SAP” or “SBP”?

6. PLOS authors have the option to publish the peer review history of their article (what does this mean?). If published, this will include your full peer review and any attached files.

Reviewer #1: No

Reviewer #2: No

---

## [Author Response · Author response to Decision Letter 0]

28 Dec 2023

Dear Dr. Wan-Jie Gu and reviewers:

We thank you for giving us such a great opportunity to revise our manuscript. All the comments are critical and helpful to improve our manuscript. We have revised the manuscript according to the comments of you and the reviewers. The revised contents have been marked in red font in the manuscript. Following are the point-by-point responses.

Editor’s comments

Response: Thank you for your advice. We have made corrections to meet PLOS ONE’s style.

Response: Thank you for your advice. The information of consent has been mentioned in section of Method as “Before answering the electronic questionnaire, consent will be informed through a cover letter”, and “Consent will be informed through a cover letter on the questionnaire homepage. Once the name of investigator was filled in the questionnaire, it is considered to obtain the informed consent” was supplied (Line 119-120). In addition, due to some translation errors, the information of consent in cover letter of questionnaire was ambiguous, thus we have corrected it as “Once you agree to answer this questionnaire with your name filled in, it will be considered as obtaining your informed consent”(S1 File).

Response: Response: Thank you for your advice. We apologize for forgetting to provide this information of Financial Disclosure section during the first submission process. However, during this revision, it seems that there is no place on the webpage where correction could be made. Thus, we have made a correction in section of “Conflict of interest statement”. We would greatly appreciate if you could help us correct the information of Financial disclosure. Besides, all the founding information has been carefully checked. (Line 342, 346-347, 350)

4. We note that Figure 1A in your submission contain [map/satellite] images which may be copyrighted. All PLOS content is published under the Creative Commons Attribution License (CC BY 4.0), which means that the manuscript, images, and Supporting Information files will be freely available online, and any third party is permitted to access, download, copy, distribute, and use these materials in any way, even commercially, with proper attribution. For these reasons, we cannot publish previously copyrighted maps or satellite images created using proprietary data, such as Google software (Google Maps, Street View, and Earth). For more information, see our copyright guidelines: http://journals.plos.org/plosone/s/licenses-and-copyright.

1) You may seek permission from the original copyright holder of Figure 1A to publish the content specifically under the CC BY 4.0 license.

2) If you are unable to obtain permission from the original copyright holder to publish these figures under the CC BY 4.0 license or if the copyright holder’s requirements are incompatible with the CC BY 4.0 license, please either i) remove the figure or ii) supply a replacement figure that complies with the CC BY 4.0 license. Please check copyright information on all replacement figures and update the figure caption with source information. If applicable, please specify in the figure caption text when a figure is similar but not identical to the original image and is therefore for illustrative purposes only.

Response: Thank you for your valuable advice. We have changed the type of Fig1A to heatmap, without changing the related data. (Fig 1A)

Response: Thank you for your suggestion. Supporting information has been added. (Line 433-457)

6. We notice that your supplementary figures are uploaded with the file type 'Figure'. Please amend the file type to 'Supporting Information'. Please ensure that each Supporting Information file has a legend listed in the manuscript after the references list.

Response: Thank you for your suggestion. Supporting information has been added, and legends have been listed. (Line 433-457)

Response: Thank you for your suggestion. We have checked the reference. 

Response to reviewers

Reviewer #1: 

This is a rare study of perioperative anesthesia management for kidney transplantation. This study is very interesting because studies in related fields are rare. The researcher collected basic information on anesthesia management for renal transplantation in China, which is helpful for Chinese anesthesiologists to enhance perioperative anesthesia management and improve the outcome of renal transplantation surgery. However, the article still needs some improvement.

1. Please elaborate on the statistical scheme.

Response: Thank you for your valuable advice. We have added the Statistical Analysis at the end of MATERIALS AND METHODS. (Line 121-126)

2. the discussion section deals with the issue of perioperative volume management, which is of interest to many clinicians. Based on the huge data support, is it possible to draw the advantages and disadvantages of perioperative volume management measures? 

Response: Thank you for comments. Perioperative volume management is an important issue for kidney transplantation. Our survey shows the current situation of perioperative volume management in China, revealing some differences between clinical practice and guideline recommendations. For example, the controversial HES is still used in 13 centers, while fluid management is guided mostly by CVP. This indicates some shortcomings in clinical practice, thus urging medical caregivers to make some improvements, and to conduct more clinical studies to explore the advantages and disadvantages of perioperative volume management measures. （Line 278-279）

3. For many perioperative management protocols, there are variations in different clinical centers, and does this variation affect the prognosis of renal transplantation?

Response: Thank you for your helpful comments. This is a good question. Our survey has investigated if there was a protocol for each center, but we have not asked which specific protocol was followed. The result showed that 84% of the investigated centres established specialized protocol (Line 138-140). However, it is unknown whether the prognosis is associated with the different protocols followed. This is a great research direction in the future. 

Reviewer #2: 

The manuscript "Perioperative management of kidney transplantation in China: A national survey in 2021" is very well written and structured, and conclusions are supported by high quality data. The importance and novelty of this study compared to published work in this subject area is good. However, there are still some issues needed to be addressed.

1 In this study, the authors applied wechat questionnaire. The response rate is high. Are there any disadvantages or limitations for this kind of electronic online survey?

Response: Thank you for your comments. For this kind of electronic survey, there are some possible disadvantages. 1) The design of the questions may have certain limitations. We have designed questions as option types instead of open-ended questions. Moreover, we conducted a pilot study before releasing the final version of questionnaire. These could reduce deficiencies in questionnaire design. However, in terms of the results, there are still certain shortcomings, such as the setting of the CVP issue in Question 28 which you mentioned. We have added this limitation in the end of Discussion. （Line 317-318）

2) There may be a possibility of duplicate responses in electronic questionnaires, so we conducted extensive data cleaning in the later stage and removed duplicates through follow-up visits.

2 The nephrotoxicity of sevoflurane has been widely reported due to the production of “compound A”. However, most of the centers routinely used sevoflurane during renal transplantaion. This point should be discussed.

Response: Thank you for your valuable advice. We have added content about this point in the Discussion section. （Line 280-285）

3 CVP can be the target of GDFT, just like SVV,PPV,or CO?

Response: Thank you for your valuable suggestions. Generally, CVP guided fluid management is considered as conventional fluid therapy (CFT). Currently, it is suggested that GDFT is a method of fluid management guided by accurate hemodynamic indicators, such as PPV, SVV, SPV, and other complex parameters. Furthermore, there are many clinical studies comparing CFT guided by CVP with GDFT[1-3]. Therefore, in questionnaire design, we distinguish between these two indicators in order to obtain more accurate results. From the results, 33 centres perform fluid therapy guided by CVP, while 30 centres selected GDFT.

4 For questionnaire, there are some typos. For example,

② Could you expand the abbreviation of MCQ ?

Response: Yes. The full name of MCQ was annotated when it first appeared in Question 12. We have made correction. （Q12 in S1 File）

②Q3,Q4: If the interviewee is not an anesthesiologist, the questionnaire was analysed differently?

Response: Thank you for your comments. In period of data cleaning, we spent a lot of effort in connecting with the chief anesthesiologist or related anesthesiologist of each kidney transplant centres in online or offline, verifying data from the replied centers, to ensure that the data was truthfully obtained from the anesthesiologists of each center. Thus, even if the direct interviewee is not an anesthesiologist, the accurate data filled in is from related anesthesiologist.

③Q15,Q17,Q36,Q42,Q44: “rocu ,[propofol,sufentanyl,fentanyl],adrenaline,[fentanyl , sufentanyl], general medical ward” . Please keep the first letter in capitals.

Response: Thank you for your valuable advice. Correspondingly, we have made these corrections. (Q15, Q17 Q36, Q42, Q44 in S1 File)

④Q18:Could you specify the monitor of depth of anesthesia?

Response: Thank you for your comments. Normally, the most commonly used monitor of depth of anesthesia is EEG monitoring in China. Considering the diversity of brands used in each center, we have not specified the types in the survey, and this was not the focus of our survey. 

⑤Q19：“Goal-directed hemodynamic therapy (GDFT)” and “CVP-guided”, there is some overlap.

Response: Thank you for your valuable advice. Recently, CVP-guided therapy is considered as conventional fluid therapy, while GDFT is supposed to be a more precise and complex monitoring method with targets of SVV, PPV, SPV. There are various studies to compare these two methods of fluid therapy, thus it seems that there is little overlap between these two. 

⑥Q28：“□<4% □4-6% □7-9% □10-12% □>12%”. If the baseline CVP is 10, the target range should be 0.4，0.4-0.6，0.7-0.9，1-1.2，1.2. However, “0.4，0.4-0.6，0.7-0.9” are not different in clinical practice.

Response: Thank you for your valuable suggestion. There is indeed certain deficiency in this Question. We have added the discussion about this deficiency in the limitation section. （Line 317-318 in manuscript）

⑦Q31: If the target is intraoperative MBP but not SBP, how to answer this question?

Response: Thank you for your valuable advice. As discussed in section of Discussion, referring to relevant guidelines in China, it is only recommended to regulate according to SBP, rather than MBP[4]. Guideline or suggestions for controlling preoperative hypertension in end-stage kidney disease are also proposed to control SBP, not MAP[5]. Currently, SBP is mostly used for targeting in clinical practice. Therefore, when designing the questionnaire, we only set questions for SBP. Furthermore, in the end of the questionnaire, we set a feedback question for interviewee. In the subsequent survey, we will further differentiate between these two.

⑧Q35: TTE is non-invasive. TEE is invasive. They are not comparable. Specify should be better.

Response: Thank you for your comments. This is indeed a very good question. The aim of Q35 is to investigate whether there is an application of ultrasound assisted method to evaluate volume and other conditions in current kidney transplantation. Thus, we have not specified invasive or non-invasive. From the results, most centres do not use TTE or TEE, with only 3 centres using TEE（Line 201-202 in Manuscript）, and this issue does not affect the conclusion. In the subsequent survey, we will further differentiate between these two.

⑨Q37：What does “Hormone” refer to? Predisone？Methyl-prednisone?Estrogen?

Response: Thank you for your valuable advice. “Hormone” refers to Glucocorticoids. There may be some translation errors, thus we have made corrections in the relevant sections of the text, charts, etc. In addition, due to the possibility of different types of glucocorticoids used in various centres, we have not specified their types. Moreover, this is not the focus of our survey. （Line 210 in Manuscript, and Q37 in S1 File）

5 For figures, the following questions should be addressed.

①Fig1: the style of figure legend (fig1C, fig1D, and fig1E) is not consistent.

Response: Thank you for your valuable advice. We have modified Fig1E to make their style consistent.

②Fig1C:“non-” N should be in capital.

Response: Thank you for your valuable advice. Correspondingly, we have made correction in Fig 1C.

③Fig1E: multiple electrolytes injection refers to sodium acetate/carbonate Ringer’s injection?

Response: Thank you for your valuable advice. Multiple electrolytes injection refers to the injection such as NORMOSOL-R solution which are widely used in clinical practice recently in China.

6 Line161 “sufentail”, line162 “remifentanyl”, line 291”SAP” or “SBP”?

Response: Thank you for your valuable advice. We have made these corrections. (Line 164-165, 292)

Reference

[1] Chirnoaga D, Coeckelenbergh S, Ickx B, et al. Impact of conventional vs. goal-directed fluid therapy on urethral tissue perfusion in patients undergoing liver surgery: A pilot randomised controlled trial [J]. European journal of anaesthesiology, 2022, 39(4): 324-32.

[2] Cavaleri M, Veroux M, Palermo F, et al. Perioperative Goal-Directed Therapy during Kidney Transplantation: An Impact Evaluation on the Major Postoperative Complications [J]. Journal of clinical medicine, 2019, 8(1): 

[3] Bloria S, Panda N, Jangra K, et al. Goal-directed Fluid Therapy Versus Conventional Fluid Therapy During Craniotomy and Clipping of Cerebral Aneurysm: A Prospective Randomized Controlled Trial [J]. Journal of neurosurgical anesthesiology, 2022, 34(4): 407-14.

[4] Chinese Society of Organ Transplantation C M A. Chinese Anesthesia technique guideline for renal transplantation(version 2019) [J]. Chinese journal of transplantation（Electronic Edition）, 2020, 14(1): 17-20.

[5] Pugh D, Gallacher P, Dhaun N. Management of Hypertension in Chronic Kidney Disease [J]. Drugs, 2019, 79(4): 365-79.

---

## [Decision Letter · Decision Letter 1]

17 Jan 2024

Perioperative management of kidney transplantation in China: A national survey in 2021

PONE-D-23-30552R1

Dear Dr. Su,

We’re pleased to inform you that your manuscript has been judged scientifically suitable for publication and will be formally accepted for publication once it meets all outstanding technical requirements.

Kind regards,

Wan-Jie Gu

Academic Editor

PLOS ONE

Additional Editor Comments (optional):

Reviewers' comments:

Reviewer's Responses to Questions

**Comments to the Author**

1. If the authors have adequately addressed your comments raised in a previous round of review and you feel that this manuscript is now acceptable for publication, you may indicate that here to bypass the “Comments to the Author” section, enter your conflict of interest statement in the “Confidential to Editor” section, and submit your "Accept" recommendation.

Reviewer #2: All comments have been addressed

2. Is the manuscript technically sound, and do the data support the conclusions?

Reviewer #2: (No Response)

3. Has the statistical analysis been performed appropriately and rigorously? 

Reviewer #2: (No Response)

4. Have the authors made all data underlying the findings in their manuscript fully available?

Reviewer #2: (No Response)

5. Is the manuscript presented in an intelligible fashion and written in standard English?

Reviewer #2: (No Response)

6. Review Comments to the Author

Reviewer #2: (No Response)

7. PLOS authors have the option to publish the peer review history of their article (what does this mean?). If published, this will include your full peer review and any attached files.

Reviewer #2: No

---

## [Editor Report · Acceptance letter]

5 Feb 2024

PONE-D-23-30552R1 

PLOS ONE

Dear Dr. Su, 

I'm pleased to inform you that your manuscript has been deemed suitable for publication in PLOS ONE. Congratulations! Your manuscript is now being handed over to our production team.

Kind regards, 

on behalf of

Dr. Wan-Jie Gu 

Academic Editor

PLOS ONE